# Risk Factors Associated with Urothelial Bladder Cancer

**DOI:** 10.3390/ijerph21070954

**Published:** 2024-07-22

**Authors:** Souhail Alouini

**Affiliations:** 1Department of Surgery, Uro-Gynecologist, Centre Hospitalier Universitaire d’ Orleans, 45100 Orleans, France; souhail.alouini@chu-orleans.fr; Tel.: +33-688395759; 2Faculté de Médecine, Université d’Orleans, 45100 Orleans, France

**Keywords:** urothelial bladder cancer, smoking, genetic, air pollution, chlorinated water, chlorinated swimming pool, occupational risks, VOCs, PAHs, particulate matter

## Abstract

Background: Urothelial bladder carcinoma (UBC) is the most frequent histologic form of bladder cancer, constituting 90% of the cases. It is important to know the risk factors of UBC to avoid them and to decrease its recurrence after treatment. The aim of this review was to provide an overview of the risk factors associated with UBC incidence. Methods: A comprehensive literature search from 2012 to 2024 was carried out in databases such as PubMed, Google Scholar, and Medline with potential keywords such as “bladder cancer”, “urothelial bladder cancer”, “incidence of urothelial bladder cancer worldwide”, “mortality rate of bladder cancer”, “incidence according to gender”, “treatment for bladder cancer”, and “risk factors of bladder cancer”. Smoking tobacco was comprehended to be the major risk factor for UBC. Smoke from tobacco products contains polycyclic aromatic hydrocarbons (PAHs) and aromatic amines such as 4-aminobiphenyl, which are known to cause UBC. Smoking-related bladder cancer mortality ranks just second to smoking-related lung cancer mortality. For non-smokers, pollution became a major risk factor associated with UBC. Polycyclic aromatic hydrocarbons (PAHs) are linked to many cancers, especially to UBC. Indoor and outdoor pollution generates VOCs (volatile organic compounds) and PAHs. Small-particle matter < 2.5 is linked to UBC and lung cancers. Drinking chlorinated water is linked to UBC. Also, swimming in chlorinated pools that produce trihalomethanes increases the risk of many cancers, and especially of bladder cancer. Occupational exposure to carcinogens, specifically aromatic amines, is a significant UBC risk factor. It has been estimated that approximately 20% of all UBCs may be linked to this type of exposure, primarily in industrial settings that treat dye, paint, petroleum chemicals, and metal. The other risk factors included genetics, diet, and medical conditions. Alcohol, consumption of processed meat and whole milk, and higher intakes of selenium and vitamins A and E also contribute to the development of UBC. Further, chemotherapeutic agents, oral hypoglycemic drugs, and radiation therapy are positively associated with UBC. Conclusions: The significance of the initial prevention of UBC must be emphasized, and especially programs for quitting cigarettes should be encouraged and supported. However, smoking is not the only risk factor for UBC. For non-smokers, other risk factors should be investigated. Air and water pollution are linked to UBC. Indoor and outdoor pollution should be more controlled. Patients and people should be informed of the risk of drinking chlorinated water and swimming in chlorinated pools.

## 1. Introduction

Bladder cancer (BC) is considered one of the most prevalent genitourinary tract malignancies [1]. Other types of bladder cancer include adenocarcinoma, small-cell carcinoma, sarcoma, and squamous-cell carcinoma, and in 90% of cases, urothelial bladder carcinoma (UBC) is the most predominant histologic form of BC [1,2]. UBC is defined as urothelial neoplastic cells invading the basement membrane, lamina propria, or the muscle. The term transitional cell carcinoma has been superseded by urothelial carcinoma by the World Health Organization (WHO) [2]. Bladder cancer is the fourth most common malignancy among men in the Western world. The rate of BC comprises between 5 and 10% of all cancers among men [3]. The median age at diagnosis is 70 years.

The mortality rates vary from 2 to 10 per 100,000 per year for men and 0.5 to 4 per 100,000 per year for women [3].

As per the rate of incidence, GLOBOCAN 2022 has ranked BC as the ninth highest. Following the same database, there were 613,799 new cases of BC, constituting 471,077 (76.7%) males and 142,722 (23.3%) females in 2022. BC was ranked 13th in mortality rates in 2022 [3]. Following the same trend of incidence, in a total of 220,347 deaths, the mortality rate was higher in males (75.1%) in 2022 [4,5,6]. The age-standardized rate (ASR) for incidence and mortality in the same year were 5.6 and 1.9, respectively [4]. The 5-year prevalence of bladder cancer in the year 2022 was 490,902, with the highest prevalence in Europe (154.4%), followed by Asia (131.1%), North America (66.8%), LAC (21.5%), Africa (19.6%), and Oceania (3.6%) [4]. In Europe, there are more tobacco smokers than in the USA. In addition, the Western diet is associated with an increased BC risk (HR: 1.54, 95% CI: 1.37–1.72) [6].

According to the former reports, the incidence of urothelial cancer is higher in males in comparison to females [7,8,9].

The pathologic severity of the disease at the time of the transurethral resection of the bladder tumor (TURBT) and subsequent staging in accordance with the tumor–node–metastasis classification system determines how bladder cancer is managed [10]. Treatment for non-muscle-invasive disease often consists of TURBT, which is followed by intravesical chemotherapy or a single dose of bacille Calmette–Guérin (BCG) immunotherapy for tumors with a higher risk of progression or recurrence [10]. Muscle-invasive bladder cancers are usually treated with radical cystectomy with extensive lymphadenectomy, followed by neoadjuvant chemotherapy based on cisplatin because of the elevated risk of progression and recurrence [10]. Cryotherapy is a minimally invasive treatment that has been used extensively for urological malignancies [11]. According to a recent report, bladder tumor recurrence can be prevented by endoscopic cryoablation, which destroys tumor cells and releases tumor antigens to foster long-lasting tumor-specific immunity. Patients with advanced bladder cancer now have additional therapy options when endoscopic cryoablation and immune-checkpoint inhibitors are combined [12]. This review article aimed to summarize the risk factors associated with the occurrence of UBC. In an attempt to draw a comprehensive picture, we tried including the major risk factors of bladder cancer. 

## 2. Methods

A comprehensive literature search was carried out in PubMed, Google Scholar, and Medline databases to obtain information for this review. The keywords used to search the previous literature were “bladder cancer”, “urothelial bladder cancer”, “incidence of urothelial bladder cancer worldwide”, “mortality rate of bladder cancer”, “incidence according to gender”, “treatment for bladder cancer”, and “risk factors of bladder cancer”. The pertinent articles between the time period 1 January 2012 to 1 January 2024 were selected. We included review articles, systematic reviews, meta-analyses, and research studies. We selected only articles published in the English language. Further, the articles were scrutinized on the basis of the relevance of the subject matter. We attempted to address the risk factors from a comprehensive perspective for the development of UBC.

### 2.1. Risk Factors for Urothelial Bladder Cancer

An extensive literature search led to the finding of several occupational and lifestyle factors that stand as major risk factors for UBC. The incidence and pathogenesis of UBC may be influenced differently by each risk factor. This phenomenon is referred to as attributable risk or the etiologic portion of risk [13]. The major risk factors for UBC are discussed below and illustrated in Table 1.

### 2.2. Smoking Tobacco

Smoking is a major risk factor for UBC, with an estimated 50% of bladder cancers resulting from it [14]. The metanalysis of Cumberbatch et al. [14] included 83 studies and showed a strong association between tobacco use and BC. The pooled relative risk for current versus never smokers was 3.47 (95% confidence interval [CI] 3.07–3.91). Tobacco is a rich source of carcinogenic compounds [15]. Tobacco smoke contains more than 60 carcinogens. Amongst them, polycyclic aromatic hydrocarbons (e.g., benzo[*a*]pyrene, nitrosamines like 4-(methylnitrosamino)-1-(3-pyridyl)-1-butanone and N′-nitrosonornicotine, aromatic amines (such as 4-aminobiphenyl), aldehydes, phenols, volatile hydrocarbons, and nitro compounds are identified [16]. 

Smoke from tobacco products contains polycyclic aromatic hydrocarbons and aromatic amines such as b-naphthylamine, which are known to cause UBC [15]. Cigarette smoking is a source of 4-aminobiphenyl (4-ABP), a recognized human bladder carcinogen [15]. Smoking cigarettes, pipes, cigars, and e-cigarettes causes inhalation of carcinogens filtered by the kidney in contact with the urinary bladder [17]. Carcinogens from tobacco smoke cause bladder and ureter cancers and renal-cell carcinoma [16,18].

These compounds cause base changes, bulky adduct formation, and double-stranded breaks in DNA [15]. The exposure of mice to PAHs and aldehydes revealed DNA adducts in bladder mucosa tissues [19]. Aldehydes are a major carcinogen that induce DNA damage and inhibit its repair [20]. In smokers, the level of methylated metabolites, such as PAHs, and aromatic amines are increased; these metabolites are responsible for smoking-attributable BC [20].

Activation of NNK and PAHs by cytochrome P450 enzymes allows the binding of these carcinogens to DNA and the formation of bulky DNA adducts [21]. 

Damaged adducts can cause gene mutations and alter the tumor suppressor genes. BC tissues of smokers showed a high level of DNA adducts of NNK and BaP contrary to non-smokers [22].

Jin et al. [23] found an elevated level of methylated metabolites of polycyclic aromatic hydrocarbons and DNA damage in smokers with bladder cancer. The treatment of BC cells with the methylation inhibitor 5-aza-2′-deoxycytidine rewired the methylated metabolites and DNA damage [16,24]. A high level of carcinogen adducts and the intensity of exposure to carcinogens may be associated with BC [25].

Also, nicotine activates nicotinic acetylcholine receptors that promote tumor progression [25]. These carcinogenic compounds are eliminated by the kidneys and hence cause cancer throughout the urinary system [14]. Smoking-related bladder cancer mortality ranks just second to smoking-related lung cancer mortality [22,26]. A meta-analysis also showed that compared to unexposed non-smokers, passive second-hand smokers have a 22% higher incidence of bladder cancer [26]. Although it is unclear if hormonal variations also play a part, the variations in smoking incidence rates across genders are commonly linked to distinct evidenced smoking trends [13]. Another potential risk factor for UBC has been linked to exposure of the environment to tobacco smoke. In general, women, especially women who had never smoked, showed the highest effects of environmental exposure to tobacco smoke. Uncertainty surrounds the precise environmental and fundamental causes of this disparity in gender [13]. Smoking duration and intensity are linked to increased risk of UBC [27,28].

### 2.3. Pollution

#### 2.3.1. Water Pollution

##### Arsenic

Arsenic. Exposure to arsenic in drinking water is a recognized cause of BC [29]. Drinking water contaminated with arsenic is a well-known risk factor for BC [29,30]. The mechanism of arsenic to induce BC remains unclear. Arsenic inhibits some sulfhydryl enzymes and antioxidant function and also has cytotoxic action [31]. Arsenic seems to cause the mutation of p53 genes in chronically exposed populations [32]; these mutations may be involved in BC.

A meta-analysis of 17 studies and a comprehensive review of 40 research studies revealed a risk effect of 5.8 (2.9–8.7) for 140 mg/L and 2.7 (95% CI 1.2–4.1) for 10 mg/L [33]. Another study reported that exposure to low concentrations of arsenic (0.5 µg/L) had a synergistic effect in increasing the risk of BC [34,35]. However, interindividual differences in arsenic metabolism may play a role in cancer risks among people exposed to low concentrations of arsenic in water [29,36].

##### Chlorinated Water

Weisman et al. [37] showed that 8000 (10%) of the 79,000 annual bladder cancer cases in the United States were potentially attributed to disinfectant products in drinking water systems.

Chlorinated water produces trihalomethane (THM). People who drank water with THM levels of more than 49 micrograms per liter had double the risk of bladder cancer compared to those who drank water with THM < 8 µg/L [38].

According to Villanueva et al. [38], participants who drank chlorinated water presented 35% greater risk of bladder cancer than those who did not.

Bladder cancer risk was associated with long-term exposure to THMs in chlorinated water at levels regularly occurring in industrialized countries [37].

In a systematic review of 1280 studies, women had higher rates of bladder cancer than men when exposed to trihalomethanes and chlorine in drinking water [38]. There are more than 700 different disinfection byproducts.

THMs are formed from a reaction between chlorinated water and organic and inorganic compounds [39]. THMs include chloroform (TCM), bromodichloromethane (BDCM), dibromochloromethane (DBCM), and bromoform (TBM) and are recognized as carcinogenic agents, especially for bladder cancer [40].

Swimming in a chlorinated pool results in high exposure levels to disinfection by products and showed genomic responses that are indicative of increased bladder risk [40].

Participants who regularly used public swimming pools boosted bladder cancer risk by 57% [37].

Swimming in pools with exposure to chlorination by-products through inhalation, ingestion, and dermal absorption is associated with bladder cancer [41], with an odds ratio of 1.57 (95% CI; 1.15–2.09).

#### 2.3.2. Air Pollution

Due to its effects on both individual and public health, as well as the escalating rates of illness and mortality it causes, air pollution is one of the major issues societies are facing today. Human disease is significantly influenced by a multitude of pollutants. One of these is particulate matter (PM), which is particles of various sizes but with incredibly small diameters of <2.5 PM that are inhaled into the respiratory system and can cause respiratory and related ailments, cancer, and problems with the cardiovascular, reproductive, and central neurological systems [42]. The following are categorized as air pollutants that are hazardous to humans: dioxins, sulfur dioxide, nitrogen oxide, polycyclic aromatic hydrocarbons (PAHs), and volatile organic compounds (VOCs). High concentrations of carbon monoxide can potentially cause immediate poisoning through inhalation. Heavy metals such as lead can intoxicate humans for a long time or induce acute poisoning, depending on the level of exposure [43]. Recently, it has been reported that there may not be a safe amount of air pollution as a carcinogen because elderly men who experience low-level ambient air pollution are more likely to develop potential bladder cancer [43]. Positive correlations between air pollution and BC risk and mortality have been found in a systematic evaluation of eight studies on the disease. A handful of them were statistically significant (e.g., an adjusted odds ratio of 1.13 (1.03–1.23) for an increase of 4.4 μg.m^−3^ of PM2.5 was associated with bladder cancer mortality) [44]. The association reached statistical significance for PM2.5 [45] and for NO_2_ [46].

Long-term exposure to polycyclic aromatic hydrocarbons (PAHs) increases the risk of bladder cancer [42].

The sources of PAHs are multiple [47].

Volatile organic compounds (VOCs)


*Indoor pollution:*
NO_2_,particulate matter (PM),gas cooking stoves,fuel,paint vapors,combustion,solvents,formaldehyde and furnishing.



*Outdoor pollution*


Diesel evaporationand petrochemical solvents are responsible for outdoor pollution and have been associated with BC. Two case-control studies (5121 patients) reported an increased risk of BC for patients exposed to high levels of diesel exhaust [48].3-nitrobenzanthrone (3-NBA) can form DNA adducts and is found in diesel exhaust. In vitro, 3-NBA is associated with higher levels of DNA adducts and DNA damage in highly undifferentiated and invasive bladder cancer cell lines compared with low-grade cell lines [49].

#### 2.3.3. Nitrate

Not much research has been performed on nitrate being a risk factor for UBC. There has been no discernible change in the outcomes of drinking water containing nitrates according to prior studies [50].

### 2.4. Genetics

The immediate members of the family of UBC patients have a twofold increased risk of developing UBC. Genetic slow acetylator N-acetyltransferase 2 (NAT2) variations and glutathione S-transferase mu 1 (GSTM1)–null genotypes are examples of inherited genetic variables that have been identified as risk factors for UBC [51]. It is well established that there is a correlation between the risk of BC and the acetylation state of NAT2 and the copy number of GSTM1 [50]. Figueroa et al. reported that three BC susceptibility variations, namely the rs798766 (TMEM129-TACC3-FGFR3), rs11892031 (UDP glycuronyltransferase 1A [UGT1A]), and GSTM1 deletion polymorphisms showed significant evidence of additive interactions [47]. Individuals with a low-activity GSTA1 genotype, as well as GSTM1- and GSTT1-active patients, were found to have an increased risk of breast cancer when exposed to solvents and pesticides (GSTT1-active) [50]. Another case-control study found that those exposed to aromatic amines and carbolineum, as well as painters and varnishers, were at a higher risk for GSTM1 and UGT1A, respectively [43]. Another case-control study found links between exposure to pesticides, genetic variants for NQO1 and SOD2, and BC risk in male agricultural workers. Participants with high-SOD2 and low-NQO1 genotypes had a higher risk of BC during activities [45]. A case-control study assessed the effect of single-nucleotide polymorphisms (SNPs) in DNA repair genes as a risk for developing UBC. They reported that the XPC PAT +/+ genotype was related to a twofold higher incidence of UBC [52]. Cowden’s syndrome is an inherited defect in the tumor-suppressor gene PTEN that can lead to several neoplasms, including urothelial bladder carcinoma [36].

### 2.5. Diet

#### 2.5.1. Alcohol

Botteri et al., from a European Prospective Investigation into Cancer and Nutrition (EPIC) cohort study, concluded that there is no established correlation between alcohol use and the risk of UBC [53]. High intake was associated to an elevated risk of UBC, although no dosage response was reported, potentially due to lifestyle factors [15].

#### 2.5.2. Meat

Processed meat consumption has been linked to bladder cancer, potentially due to various enzymatic, mechanical, and chemical treatments. Nitrite, a color and flavor preservative used in processed meat, could interact with secondary amines from proteins to generate nitrosamines, which have been linked to bladder cancer, among other cancers [54,55]. In two case-control studies, processed meats and animal protein were linked to breast cancer [56,57]. A 22% increased risk for BC by processed meats has been reported previously, but there is no statistically significant increase for red meats [36]. However, another meta-analysis reported that a high intake of processed and red meat increased bladder cancer risk by 10% and 17%, respectively [36]. On the contrary, consuming pork and poultry is not correlated with a higher risk of BC [36].

#### 2.5.3. Fruits and Vegetables

Previous studies have reported that consuming fruits and vegetables is related to a lower risk of bladder cancer. The same was observed for citrus fruits and cruciferous vegetables [57]. The EPIC study, which included nearly 500,000 adults, showed a lack of connection between vegetable and fruit intake and UBC risk. The risk of BC was marginally significant when comparing the greatest tertile with the lowest tertile of combined fruit and vegetable consumption (HR:1.30; 95% CI, 1.00–1.69) [14]. In contrast, a meta-analysis of 15 prospective cohort studies found a summary risk rate of 0.97 (95% CI 0.95–0.99) for an increase in one serving per day of fruits and vegetables [58]. However, there are other reports suggesting that low fruit and vegetable intake, as well as urban living, is not always related to an increased risk of BC [59,60].

#### 2.5.4. Vitamins

The previous literature suggests that higher intakes of selenium, vitamins A and E, and folate, but not vitamin C, were observed to significantly reduce the incidence of bladder cancer, with effects of −39%, −18%, and 16%, respectively [55]. In contrast, another study showed a 52% rise in the risk of bladder cancer among participants receiving supplements in experimental investigations. These inconsistent findings could be due to a dose-dependent interaction, with ingestion being protective at low concentrations and detrimental at large doses [55]. A prospective series of roughly 80,000 people evaluated the effect of prolonged usage of supplemental minerals and vitamins on the risk of UBC; after a 6-year average follow-up, no supplement was shown to be substantially associated in multivariate models [13]. A recent meta-analysis by Gu et al. has reported that increasing folate consumption to 100 μg/day reduced the incidence of invasive UC by 8% (RR = 0.92, 95% CI: 0.87–0.98, *p* = 0.004). They further predicted that increased folate intake could be inversely related to the risk of UC, especially invasive UC [27]. In contrast, a meta-analysis of seven studies (two cohorts and five case-control) found a pooled risk ratio (RR) of 1.34 (95% CI 1.17–1.53) for low versus high vitamin D levels [35]. Moreover, a meta-analysis of five studies—two cohorts and three case-control—showed that a high serum vitamin D level protected against BC risk (RR 0.75, 95% CI 0.65–0.87) [35]

#### 2.5.5. Dairy

A significant decrease in the incidence of BC has been observed in relation to milk, fermented milk, and skim milk (milk: RR 0.84; 95% CI 0.72–0.97; I2 = 70.1%; *n* = 16; fermented milk: RR 0.69; 95% CI 0.47–0.91; I2 = 62.5; *n* = 5; skim milk: RR 0.47; 95% CI 0.18–0.79; I2 = 0; *n* = 2). On the other hand, drinking whole milk was linked to a significantly higher risk of BC [55].

### 2.6. Medical Conditions

Individuals may be susceptible to UBC as a direct result of their medical issues or due to toxicity of their treatment. Two instances of direct causal roles include carcinogenesis linked to chronic inflammation or schistosomiasis and carcinogenesis connected with prolonged urine retention and upper tract dilatation, increasing urothelial exposure to carcinogens [45]. A higher ASR incidence rate of UBC followed by external-beam radiation for prostate cancer has been reported (HR: 1.70; 95% CI, 1.57–1.86) [55]. Cyclophosphamide, a chemotherapeutic agent used in the treatment of leukemia and lymphoma, is suggested to be associated with a sustained rise in the incidence of UBC [55,61]. Cyclophosphamide produces the metabolite acrolein, which is associated with hemorrhagic cystitis and mutagenesis of the urinary bladder. A cumulative, dose-dependent exposure to this agent increase the risk of bladder cancer [62].

There has been evidence of a higher incidence of UBC for diabetes mellitus, and this incidence increased with a longer period and consumption of oral hypoglycemic drugs (HR: 2.2; 95% CI, 1.3–3.8) [55]. A meta-analysis of 36 studies (nine case-control and 27 cohort) revealed a negative correlation between the duration of diabetes mellitus and the risk rate of BC, with patients diagnosed with the disease for less than five years having a higher risk of BC [62]. For obesity and overweight, respectively, Sun et al. showed pooled risk rates of BC of 1.10 (1.06–1.14) and 1.07 (95% CI 1.01–1.14) [62]. Previously, it was observed that among non-smokers (RR 0.57, 95% CI 0.43–0.76) but not smokers currently in the habit, the intake of non-aspirin, non-steroidal anti-inflammatory drug therapy was substantially linked to a 43% reduction in BC risk [17].

### 2.7. Occupational Risk

It is predicted that 7.1% of male instances of UBC can be linked to occupational variables, as men are often more exposed to these risk factors [55,63]. Textile dyers, painters, miners, and dry cleaners are examples of occupational groups associated with increased risk for BC [64].

The second most significant UBC risk factor, after smoking, is thought to be occupational exposure to carcinogens, specifically aromatic amines (2-naphthylamine, benzidine, 4-chloro-o-toluidine, 4-aminobiphenyl), chlorinated polycyclic aromatic hydrocarbons, metal working fluids, diesel exhaust, and perchloroethylene [13,14]. These sources may induce 4-ABP or 2,6-dimethylaniline adduct formation. There is an association between 2,6-DMA-Hb and BC risk [64,65]. Indeed, DNA adducts of 2,6-DMA can be found in the DNA of exfoliated urothelial cells in smokers and non-smokers [66].

It has been estimated that approximately 20% of all UBCs may be linked to this type of exposure, primarily in industrial settings that treat dye, paint, petroleum chemicals, and metal [14]. A case-control study reported that men operating machines in the printing industry had a statistically significant higher risk than male farmers, and after controlling for the smoking duration, the occupation had no significant correlations with BC risk among women (HR: 5.4; 96% CI, 1.6–17.7) [14]. Similarly, there was a significantly lower risk of bladder cancer for those who work in forestry, farming, gardening, teaching, and other occupations where there is probably less frequent exposure to harmful environmental agents at work and in the surrounding area [36,63].

### 2.8. Race

Additional historical Surveillance Epidemiology and End Results (SEER) data from the United States indicated that the incidence of BC in black people is higher than the mortality rate in white people [65]. African Americans have half of the risk of white Americans to develop BC. However, the reasons for this are not determined [13].

### 2.9. Gender

The preceding research indicates that women experience a higher death rate and a lower incidence of UBC than men [13]. The gender-based prevalence of UBC is described in Table 1. The gender differences in BC incidence and survival are thought to be caused by a few (probably combined) factors, including differences in access to healthcare, delayed diagnosis (hematuria or fewer symptoms of UTI causing cystitis in women), exposures related to occupation, and smoking habits [17]. Differential metabolic detox of carcinogens and variations in the sex steroid hormone pathways are examples of potential molecular processes [57]. Bladder cancer is three to four times more common in men than in women. Smoking in men is not the only factor that explains this fact. Hormonal status seems to play a role. Indeed, multiparous women seem to have a lower risk of BC than nulliparous, probably due to hormonal changes during pregnancy [3].

Moreover, some animal experiments on rats suggested that androgenic hormones stimulate bladder oncogenesis conversely to estrogenic hormones [66].

According to reports, in patients with pT1 UBC receiving therapy, the gender of the patient is an adverse predictor of time to progression, recurrence, and survival specific to cancer (death due to UBC HR: 3.53; *p* = 0.004) [13]. Another study found that women had a lower cancer-specific survival rate than males after a cystectomy for muscle-invasive UBC but that the findings improved during treatment (HR: 1.35; *p* = 0.048) [13].

### 2.10. Socio-Economic Status

In a review of patients seeking social welfare medical aid, low socioeconomic status was linked to worse UBC-specific survival; this relationship was not found in earlier findings [13].

**Table 1 ijerph-21-00954-t001:** Risk factors with the respective causative agents responsible for Urothelial Bladder Cancer.

Authors	Risk Factors for Urothelial Bladder Cancer	Causative Agents
Burger et al., 2013 [13]	Smoking	Polycyclic aromatic hydrocarbons and aromatic amines such as b-naphthylamine
Cumberbatch et al., 2018 [17]

Pesch et al., 2013 [65]
Cumberbatch et al., 2018 [17]	Pollution	Arsenic
Baris et al., 2016 [27]
Mendez et al., 2017 [34]

Cumberbatch et al., 2018 [17]	Nitrate
Manisalidis et al., 2020 [42]	Air Pollution
Lim et al., 2023 [43]
Sakhvidi et al., 2020 [44]
Weissman et al, 2022 [37]	Trihalomethane THM	Chlorinated water
Villanueva et al., 2007 [38]	THM	Chlorinated swimming pools
Burger et al., 2013 [13]	Genetics	NAT2, GSTM1, rs798766 (TMEM129-TACC3-FGFR3), rs11892031 (UDP glucuronyltransferase 1A [UGT1A]), GSTM1 GSTA1, GSTT1, NQO1, and SOD2,
Cumberbatch et al., 2018 [17]
Figueroa et al., 2015 [48]
Amr et al., 2015 [51]
Samara et al., 2021 [52]
Saginala et al., 2020 [36]
Cumberbatch et al., 2018 [17]	Diet	Alcohol
Botteri et al., 2017 [53]
Saginala et al., 2020 [36]	Meat
Al-Zalabani et al., 2016 [54]
Catsburg et al., 2014 [55]
Ronco et al., 2014 [56]
Burger et al., 2013 [13]	Fruits and Vegetables
Cumberbatch et al., 2018 [17]
Zalabani et al., 2016 [54]
Vieira et al., 2015 [57]
Sanli et al., 2017 [58]
Gu et al., 2022 [59]	Vitamins
Liao et al., 2015 [35]
Zalabani et al., 2016 [54]	Diary
Zalabani et al., 2016 [54]	Medical Conditions	Oral hypoglycemic drugs, chemotherapeutic agents, and radiation
Zhu et al., 2013 [60]
Sun et al., 2015 [62]; Cumberbatch et al., 2018 [17]
Burger et al., 2013 [13]	Occupational Risk	2-naphthylamine, benzidine, 4-chloro-o-toluidine, 4-aminobiphenyl), chlorinated, polycyclic aromatic hydrocarbons, metal working fluids, diesel exhaust, and perchloroethylene
Cumberbatch et al., 2018 [17]
Saginala et al., 2020 [36]
Cumberbatch et al., 2015 [63]
Pesch et al., 2013 [65]	Race	-
Burger et al., 2013 [13]	Gender	Male
Cumberbatch et al., 2018 [17]
	Female

## 3. Conclusions

Urothelial bladder cancer is one of the prevalent cancers in men. Though several risk factors for UBC have been identified, smoking is the most frequent. Thus, the significance of initial prevention must be emphasized, and programs for quitting cigarettes should be encouraged and supported. However, smoking is not the only risk factor for UBC. For non-smokers, other risk factors should be investigated. Air and water pollution are linked to UBC. Indoor and outdoor pollution should be more controlled. Patients and people should be informed of the risk of drinking chlorinated water and swimming in chlorinated pools.

## Data Availability

No new data were created or analyzed in this study. Data sharing is not applicable to this article.

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
