# Peer review of "Risk Factors Associated with Urothelial Bladder Cancer"

_ijerph, 2024, doi:10.3390/ijerph21070954_

Round 1

Reviewer 1 Report

Comments and Suggestions for Authors

This is a weak attempt to analyse the risk factors of a condition for which many studies and reviews on risk factrors already exist.  Nevertheless, I have presented comments/ suggestions to make the work stronger. 

Comments on the Quality of English Language

Author Response

Thank you for comments

General comments

  • Half of the introduction has concentrated on management of bladder cancer, when the study is not on management of cancer but risk factors. Answer : This part of introduction was removed in a major part.

  • Structure section 3.1 to present a more comprehensive in-depth analysis by first presenting literature on the causative agents in tobacco smoke, with multiple references. Then, present the toxicology of these agents to show they cause bladder cancer, with mechanisms of action, first in animals or cells then in humans. Present many references to back your assertion or association. Then provide more backing with the epidemiological evidence in all types of epidemiological study designs. The study is on incidence, and apparently prevalence, but there are not many incidence or prevalence figures provided.

Answer: many references are added and the sections 3-1 , 32.2 and 3.3 are improved , please see the modifications on the manuscript and see the comments below.

The same comment can be said about Sections 3.2, 3.3 and 3.4. As it is, the analysis in all these sections is weak and insufficient.

Answer: Following your suggestions, many references are added and the sections 3-1 , 32.2 and 3.3 are improved  as well, please see the modifications in the manuscript.

Specific comments

  • Lines 9 and 10: “It is important to know risk factors of UBC to avoid them and to decrease its recurrence after treatment.” The article is focussing on occurrence and not recurrence. Are the risk factors for occurrence the same as those for recurrence? Answer: Yes. It is known that stopping tabacco which is the most important RF for UBC after ablation of vesical tumor decreases the incidence of recurrence. As the recurrence after TURBT is high, therefore, it is important to know these RF and to avoid them.

reference

  • Lines 22-23: “Also, swimming in chlorinated pools produces trihalomethanes that significantly increase the risk of many cancers…”. Surely, it is not the swimming that produces THMs but chlorination of water. Answer: Yes. Replaced by: swimming in chlorinated pools that produce trihalomethanes increases the risk of many cancers…
  • Prevalence was used among the search words when the title is focusing on incidence. Yes. This was a mistake: prevalence was replaced by incidence.
  • Line 96: “An extensive literature search led to the finding of several occupational and lifestyle……” No sociodemographic factors? In methods « risks factors of UBC » include socio economic and demographic factors. Please also see: results where socio economic factors, etc.. are described.

Editorial comments: Line 21: door and outdoor pollution generates COV and HAP. Small particle matter PM < 2.5 are linked….. First time write in full   OK: replaced by volatile organic compounds, polycyclic aromatic hydrocarbons. Small particle matter, etc.

2

  • Line 22: Drinking chlorinated water areis linked to UBC. Ok, corrected.
  • Lines 102-103: “Smoking is a major risk factor for UBC, with an estimated 50% of tumors resulting from smoking”. Reference needed. ANSWER reference added

M.G. Cumberbatch, M. Rota, J.W. Catto, C. La Vecchia

The role of tobacco smoke in bladder and kidney carcinogenesis: a comparison of exposures and meta-analysis of incidence and mortality risks

Eur Urol, 70 (2016), pp. 458-466

  • Lines 104-105: “Cigarette smoking is a source of 4-aminobiphenyl (4-ABP), a recognized human bladder carcinogen”. Reference needed
  • Lines 106-107: “These compounds cause base changes, bulky adduct formation, and double-stranded breaks in DNA”. Spend more time on the toxicology of the causative agents, providing more references.
  • Line 107: “These compounds are eliminated by the kidneys and hence cause cancer throughout the urinary system”. References
  • Line 103: “Another potential risk factor for UBC has been linked to exposure of the environment to tobacco smoke”. References needed.

Answer: ok reference added in the manuscript in section 3 and references.

[150]. M.G. Cumberbatch, M. Rota, J.W. Catto, C. La Vecchia

The role of tobacco smoke in bladder and kidney carcinogenesis: a comparison of exposures and meta-analysis of incidence and mortality risks.

Eur Urol, 70 (2016), pp. 458-466

[160] M.C. Stern, J. Lin, J.D. Figueroa, et al.

Polymorphisms in DNA repair genes, smoking, and bladder cancer risk: findings from the International Consortium of Bladder Cancer

Cancer Res, 69 (2009), pp. 6857-6864

[170]. U. Gabriel, L. Li, C. Bolenz, et al.

New insights into the influence of cigarette smoking on urothelial carcinogenesis: smoking-induced gene expression in tumor-free urothelium might discriminate muscle-invasive from nonmuscle-invasive urothelial bladder cancer

Mol Carcinog, 51 (2012), pp. 907-915

[180]. J. Ferrís, O. Berbel, J. Alonso-López, J. Garcia, J.A. Ortega

Environmental non-occupational risk factors associated with bladder cancer

Acta Urol Esp, 37 (2013), pp. 579-586

 [190]. J. Polesel, C. Bosetti, M. di Maso, et al.

Duration and intensity of tobacco smoking and the risk of papillary and non-papillary transitional cell carcinoma of the bladder

Cancer Causes Control, 25 (2014), pp. 1151-1158

  1. Jin F, Thaiparambil J, Donepudi SR, Vantaku V, Piyarathna DWB, Maity S, Krishnapuram R, Putluri V, Gu F, Purwaha P, Bhowmik SK, Ambati CR, von Rundstedt FC, Roghmann F, Berg S, Noldus J, Rajapakshe K, Gödde D, Roth S, Störkel S, Degener S, Michailidis G, Kaipparettu BA, Karanam B, Terris MK, Kavuri SM, Lerner SP, Kheradmand F, Coarfa C, Sreekumar A, Lotan Y, El-Zein R, Putluri N. Tobacco-Specific Carcinogens Induce Hypermethylation, DNA Adducts, and DNA Damage in Bladder Cancer. Cancer Prev Res (Phila). 2017 Oct;10(10):588-597. doi: 10.1158/1940-6207.CAPR-17-0198. Epub 2017 Aug 29. PMID: 28851690; PMCID: PMC5626664.
  2. Radespiel-Tröger M, Geiss K, Twardella D, Maier W, Meyer M. Cancer incidence in urban, rural, and densely populated districts close to core cities in Bavaria, Germany. Int Arch Occup Environ Health. 2018 Feb;91(2):155-174. doi: 10.1007/s00420-017-1266-3. Epub 2017 Oct 12. PMID: 29027001.

  1. Collarile P, Bidoli E, Barbone F, Zanier L, Del Zotto S, Fuser S, Stel F, Panato C, Gallai I, Serraino D. Residence in Proximity of a Coal-Oil-Fired Thermal Power Plant and Risk of Lung and Bladder Cancer in North-Eastern Italy. A Population-Based Study: 1995-2009. Int J Environ Res Public Health. 2017 Jul 31;14(8):860. doi: 10.3390/ijerph14080860. PMID: 28788106; PMCID: PMC5580564.

Reviewer 2 Report

Comments and Suggestions for Authors

1. Page 1 of 12

Line number: 21

Please spell out COV and HAP.

2. Page 2 of 12

Line number: 53

The author may provide potential reasons or assumptions as to why Europe has the highest prevalence, if possible.

3. Page 2 of 12

Line number: 85

Methods

The author may provide more details regarding the methods used in this review study. For example, inclusion and exclusion criteria could be elaborated upon. Additionally, including a flow chart to present the selection process of articles or studies would allow readers to have a better visualization of the article selection process.

4. Page 7 of 12

Line number: 310-313

Table 1 and Table 2

The author may need to consider recreating these two tables that would be more appropriate for the review study. For example, including measures of association, target population, primary outcome, exposure, time, and study design for each research or paper used in your review.

5. Page 9 of 12

Line number: 315

Conclusion

It's a little difficult to fully agree with the authors' conclusions based on the current review study. Authors may need to consider providing more details or evidence used in the review study to strengthen the conclusions.

Comments on the Quality of English Language

English is good. 

Author Response

Thank you for your cpmments

Line number: 21

Please spell out COV and HAP. : Yes, mistake : replaced by VOCs (Volatile organic compounds) and PAHs

  1. Page 2 of 12

Line number: 53

The author may provide potential reasons or assumptions as to why Europe has the highest prevalence, if possible.

Added:     In Europe there are more tobacco smokers than in USA, in addition western diet is associated with an increased BC risk (HR : 1.54, 95% CI: 1.37–1.72) (100)

  1. Witlox W.J.A.,van Osch F.H.M.,Brinkman M.et al.An inverse association between the Mediterranean diet and bladder cancer risk: a pooled analysis of 13 cohort studies.Eur J Nutr. 2020; 59: 287-296

  1. Page 2 of 12

Line number: 85

Methods

The author may provide more details regarding the methods used in this review study. For example, inclusion and exclusion criteria could be elaborated upon. Additionally, including a flow chart to present the selection process of articles or studies would allow readers to have a better visualization of the article selection process.

 We included review articles, systematic review, metaanalysis and research studies. We selected only articles published in English language. According to key words in methods we searched this type of papers in the different databases. Flow chart is usually done for a systematic review. In our case it is a narrative review.

  1. Page 7 of 12

Line number: 310-313

Table 1 and Table 2

The author may need to consider recreating these two tables that would be more appropriate for the review study. For example, including measures of association, target population, primary outcome, exposure, time, and study design for each research or paper used in your review.

Answer: Table 1 is removed. A new table is recreated. As most of the papers are reviews, therefore it is not possible to include target population.

  1. Page 9 of 12

Line number: 315

Conclusion

It's a little difficult to fully agree with the authors' conclusions based on the current review study. Authors may need to consider providing more details or evidence used in the review study to strengthen the conclusions. Yes conclusion is revised and improved

added: « However, smoking is not the only risk factor for UBC. For non-smokers other risk factors should be investigated. Air and water pollution are linked to UBC. Indoor and outdoor pollution should be more controlled. Patients and people should be informed on the risk of drinking chlorinated water and swimming in chlorinated pools ».

Reviewer 3 Report

Comments and Suggestions for Authors

The authors have conducted a narrative review of risk factors for urothelial bladder cancer (UBC), including environmental factors, behavioral factors, occupational factors, genetic factors, and sociodemographic factors. As expected, the authors found that smoking was the biggest risk factor for UBC, with air pollution being another major driving agent (among non-smokers). While there is some useful information in this review, it does not provide much synthesis or depth within its comparisons of studies for a given risk factor, and there is limited information provided for any given study included in the review, particularly as it relates to measures of the exposure/risk factor. Further, some sections lack structure (e.g., lines 126-143, lines 165-173). Overall, this would require a significant rewrite to be considered for publication. 

Comments on the Quality of English Language

Minor to moderate grammatical or syntactical errors throughout. 

Author Response

Answers: Lines 126-143 , 165-173 were improved, please see answers to reviewers 1  and 2  modifications are in the manuscript , more explanations and references are added.

Comments on the Quality of English Language

Minor to moderate grammatical or syntactical errors throughout. Answer: English language was revised.

Thank you for your comments

Round 2

Reviewer 1 Report

Comments and Suggestions for Authors

I have hesiatted to give a review beacuse the cahanges I asked to 3.1, 3.2, 3.3 etc were not done sufficiently. I had asked for the toxicology of these agents to show they cause bladder cancer, with mechanisms of action, first in animals or cells then in humans. Then, present many references to back your assertion or association. Then provide more backing with the epidemiological evidence in all types of epidemiological study designs. The study is on incidence, and apparently prevalence, but there are not many incidence or prevalence figures provided. 

Author Response

Dear Editor

Thank you for your comments.

The mechanisms of oncongenesis are explained in depth according to your suggestions in this second revision. Please find below these modifications concerning sections indicated and the new references. All these modifications were introduced in the manuscript.

The “incidence “was removed from the title. The review is focused on the risk factors of urothelial bladder cancer. A paragraph concerning epidemiology was added in the introduction and in results.

Please find the new revisions in addition to previous revisions below and added in the manuscript.

Best regards.

-

3.1

Tobacco smoke contains more than 60 carcinogens. Amongst them, polycyclic aromatic hydrocarbons (e.g., benzo[a]pyrene , nitrosamines  like 4-(methylnitrosamino)-1-(3-pyridyl)-1-butanone and N′-nitrosonornicotine , aromatic amines (such as 4-aminobiphenyl), aldehydes, phenols, volatile hydrocarbons, nitro

Smoking Cigarette, pipes cigars, E cigarettes causes inhalation of carcinogens filtered by the kidney in contact with the urinary bladder (1000). Carcinogens from tobacco smoke cause bladder and ureter cancers and renal-cell carcinoma [15,18].

The exposure of mice to PAHs and aldehydes revealed DNA adducts in bladder-mucosa tissues [33]. Aldehydes are a major carcinogen that induce DNA damage and inhibit its repair [33].

In smokers the level of methylated metabolites, such as PAHs, and  aromatic amines are increased            , these metabolites are responsible for smoking attributable BC [[19].

Activation of NNK and PAHs by cytochrome P450 enzymes allows the binding of these carcinogens to DNA and the formation of bulky DNA adducts [22].

Damaged adducts can cause gene mutations and alter the tumor suppressor genes.  BC tissues of smokers showed a high level of DNA adduct of NNK and BaP

contrary in non-smokers[8].

A high level of carcinogen-adducts and  the intensity of exposure to carcinogens may be associated with BC[23].

Also, nicotine activates nicotinic acetylcholine receptors that promote tumor progression [24].

References

  1. Xue J, Yang S, Seng S. Mechanisms of Cancer Induction by Tobacco-Specific NNK and NNN. Cancers (Basel). 2014 May 14;6(2):1138-56. doi: 10.3390/cancers6021138. PMID: 24830349; PMCID: PMC4074821.
  2. Gaffney CD, Katims A, D'Souza N, Bjurlin MA, Matulewicz RS. Bladder Cancer Carcinogens: Opportunities for Risk Reduction. Eur Urol Focus. 2023 Jul;9(4):575-578. doi: 10.1016/j.euf.2023.03.017. Epub 2023 Apr 5. PMID: 37028984; PMCID: PMC10524287

  1. Chappell G, Pogribny IP, Guyton KZ, Rusyn I. Epigenetic alterations induced by genotoxic occupational and environmental human chemical carcinogens: A systematic literature review. Mutat Res Rev Mutat Res. 2016 Apr-Jun;768:27-45. doi: 10.1016/j.mrrev.2016.03.004. Epub 2016 Mar 31. PMID: 27234561; PMCID: PMC4884606.
  2. Lee HW, Wang HT, Weng MW, Chin C, Huang W, Lepor H, Wu XR, Rom WN, Chen LC, Tang MS. Cigarette side-stream smoke lung and bladder carcinogenesis: inducing mutagenic acrolein-DNA adducts, inhibiting DNA repair and enhancing anchorage-independent-growth cell transformation. Oncotarget. 2015 Oct 20;6(32):33226-36. doi: 10.18632/oncotarget.5429. PMID: 26431382; PMCID: PMC474176119. Vineis P, Pirastu R. Aromatic amines and cancer. Cancer Causes Control. 1997 May;8(3):346-55. doi: 10.1023/a:1018453104303. PMID: 9498898.

  1. . Sturla SJ, Scott J, Lao Y, Hecht SS, Villalta PW. Mass spectrometric analysis of relative levels of pyridyloxobutylation adducts formed in the reaction of DNA with a chemically activated form of the tobacco-specific carcinogen 4-(methylnitrosamino)-1-(3-pyridyl)-1-butanone. Chem Res Toxicol. 2005 Jun;18(6):1048-55. doi: 10.1021/tx050028u. PMID: 15962940.
  2. Expression of Concern: Tobacco-Specific Carcinogens Induce Hypermethylation, DNA Adducts, and DNA Damage in Bladder Cancer. Cancer Prev Res (Phila). 2019 Oct;12(10):733-734. doi: 10.1158/1940-6207.CAPR-19-0406. Epub 2019 Sep 3. PMID: 31481538.

  1. Wiencke JK. DNA adduct burden and tobacco carcinogenesis. Oncogene. 2002 Oct 21;21(48):7376-91. doi: 10.1038/sj.onc.1205799. PMID: 12379880.
  2. Warren GW, Singh AK. Nicotine and lung cancer. J Carcinog. 2013 Jan 31;12:1. doi: 10.4103/1477-3163.106680. PMID: 23599683; PMCID: PMC3622363.

Arsenic in drinking water

The mechanism of arsenic to induce BC remains unclear. Arsenic inhibits some sulfhydryl enzymes, antioxidant function and has also a cytotoxicity action [32]. Arsenic seems to cause mutation of p53 genes in chronically exposed populations (30); these mutations maybe involved in BC.

references

  1. Anetor JI, Wanibuchi H, Fukushima S: Arsenic exposure and its health effects and risk of cancer in developing countries: micronutrients as host defence. Asian Pac J Cancer Prev. 2007, 8: 13-23.

  1. Tapio S, Grosche B: Arsenic in the aetiology of cancer. Review. Mutation Research. 2006, 612: 215-146. 10.1016/j.mrrev.2006.02.001.

Diesel

Air pollution

The 3-nitrobenzanthrone (3-NBA) can form DNA adducts and is found in diesel exhaust. The  3-NBA iss associated with higher levels of DNA adducts, and DNA damage in highly  undifferentiated and invasive bladder cancer cell line  compared with low-grade  cell line [47].

  1. Reshetnikova G, Sidorenko VS, Whyard T, Lukin M, Waltzer W, Takamura-Enye T, Romanov V. Genotoxic and cytotoxic effects of the environmental pollutant 3-nitrobenzanthrone on bladder cancer cells. Exp Cell Res. 2016 Nov 15;349(1):101-108. doi: 10.1016/j.yexcr.2016.10.003. Epub 2016 Oct 5. PMID: 27720671

Cyclophosphamide

Cyclophosphamide produces the metabolite acrolein, which is associated with hemorrhagic cystitis and mutagenesis of the urinary bladder.  A  cumulative, dose-dependent exposure to this agent increase the risk of bladder cancer  [62].

  1. Travis LB, Curtis RE, Glimelius B, Holowaty EJ, Van Leeuwen FE, Lynch CF, Hagenbeek A, Stovall M, Banks PM, Adami J, et al. Bladder and kidney cancer following cyclophosphamide therapy for non-Hodgkin's lymphoma. J Natl Cancer Inst. 1995 Apr 5;87(7):524-30. doi: 10.1093/jnci/87.7.524. PMID: 7707439

Occupational risks

Textile dyers, painters, miners, dry cleaners are examples of occupational groups associated with increased risk for this cancer [[40], [41], [42], [43]]. Moreover, entire populations  are exposed to additional factors like pesticides in food or hair dye products [30]. Aromatic amines are also found in industries such as rubber, textile manufacturing, aluminum transformation, as well as gas, coal, pesticides, and cosmetics production [30].

These sources may induce 4-Aminobiphenyl or 2,6-dimethylaniline adducts formation. There is an association between 2,6-DMA-Hb and BC risk [65]. Indeed, DNA adducts of 2,6-DMA can be found in the DNA of exfoliated urothelial cells in smokers and non-smokers [66].

references

  1. Tao L, Day BW, Hu B, Xiang YB, Wang R, Stern MC, Gago-Dominguez M, Cortessis VK, Conti DV, Van Den Berg D, Pike MC, Gao YT, Yu MC, Yuan JM. Elevated 4-aminobiphenyl and 2,6-dimethylaniline hemoglobin adducts and increased risk of bladder cancer among lifelong nonsmokers--The Shanghai Bladder Cancer Study. Cancer Epidemiol Biomarkers Prev. 2013 May;22(5):937-45. doi: 10.1158/1055-9965.EPI-12-1447. PMID: 23539508; PMCID: PMC406579

  1. Reshetnikova G, Sidorenko VS, Whyard T, Lukin M, Waltzer W, Takamura-Enye T, Romanov V. Genotoxic and cytotoxic effects of the environmental pollutant 3-nitrobenzanthrone on bladder cancer cells. Exp Cell Res. 2016 Nov 15;349(1):101-108. doi: 10.1016/j.yexcr.2016.10.003. Epub 2016 Oct 5. PMID: 27720671.

.

 Bladder cancer is the fourth most common malignancy among men in the Western world. The rate of BC is comprised between 5 and 10 % of all cancer among men (68) The median age at diagnosis is 70 years.

The mortality rates vary from 2 to 10 per 100,000 per year for men and 0.5 to 4 per 100,000 per year for women (12)

African Americans have the half of the risk of White Americans to develop a BC. However, the reasons for this are not determined.

Bladder cancer is 3 to 4 times more common in men than in women. Smoking in men is not the only factor that explains this fact. Hormonal status seems to play a role. Indeed, multiparous women seem to have a lower risk of BC than nulliparous, probably due to hormonal changes during pregnancy (ref Plesko)

Moreover, some animal experiments on rats suggested that androgenic hormones stimulate bladder oncongenesis conversely to estrogenic hormones (69)

References

  1. Kirkali Z, Chan T, Manoharan M, Algaba F, Busch C, Cheng L, Kiemeney L, Kriegmair M, Montironi R, Murphy WM, Sesterhenn IA, Tachibana M, Weider J. Bladder cancer: epidemiology, staging and grading, and diagnosis. Urology. 2005 Dec;66(6 Suppl 1):4-34. doi: 10.1016/j.urology.2005.07.062. PMID: 16399414
  2. Reid LM, Leav I, Kwan PW, Russell P, Merk FB. Characterization of a human, sex steroid-responsive transitional cell carcinoma maintained as a tumor line (R198) in athymic nude mice. Cancer Res. 1984 Oct;44(10):4560-73. PMID: 6467211.

Plesko I, Preston-Martin S, Day NE, Tzonou A, Dimitrova E, Somogyi J. Parity and cancer risk in Slovakia. Int J Cancer. 1985 Nov 15;36(5):529-33. doi: 10.1002/ijc.2910360502. PMID: 4055127.

13.Burger M, Catto JW, Dalbagni G, Grossman HB, Herr H, Karakiewicz P, Kassouf W, Kiemeney LA, La Vecchia C, Shariat S, Lotan Y. Epidemiology and risk factors of urothelial bladder cancer. Eur Urol. 2013 Feb;63(2):234-41. doi: 10.1016/j.eururo.2012.07.033.

Reviewer 2 Report

Comments and Suggestions for Authors

All comments have been appropriately addressed.

Author Response

Dear Reviewer,

Thank you very much for your comments.

Best regards

Souhail Alouini

Reviewer 3 Report

Comments and Suggestions for Authors

I thank the authors for their responses to reviewer comments and concerns.

Author Response

(The authors gave the same response as above.)
